

# Associations between the rs5498 (A > G) and rs281432 (C > G) polymorphisms of the *ICAM1* gene and atherosclerotic cardiovascular disease risk, including hypercholesterolemia

Naruemon Wechjakwen[1], Amornrat Aroonnual[1], Pattaneeya Prangthip[1], Ngamphol Soonthornworasiri[2], Pornpimol Panprathip Phienluphon[1], Jirayu Lainampetch[3] and Karunee Kwanbunjan[1]

[1] Department of Tropical Nutrition and Food Science, Faculty of Tropical Medicine, Mahidol University, Bangkok, Thailand
[2] Department of Tropical Hygiene, Faculty of Tropical Medicine, Mahidol University, Bangkok, Thailand
[3] Department of Nutrition, Faculty of Public health, Mahidol University, Bangkok, Thailand

Corresponding author
Karunee Kwanbunjan,
karunee.kwa@mahidol.ac.th

## ABSTRACT

**Background.** Atherosclerotic cardiovascular disease (ASCVD) originates from complex risk factors, including age, gender, dyslipidemia, obesity, race, genetic and genetic variation. *ICAM1* gene polymorphisms are a significant risk factor for ASCVD. However, the impact of the rs5498 and rs281432 polymorphisms on the prevalence of hypercholesterolemia (HCL) has not been reported. Therefore, we determine the relationships between single nucleotide polymorphisms (SNPs), including rs5498 and rs281432 on Intercellular adhesion molecule 1 gene (*ICAM1*) and ASCVD susceptibility in patients with HCL.

**Methods.** The clinical characteristics of 278 participants were assessed, and classified to groups having HCL and without HCL. *ICAM1* SNPs genotyping was performed by DNA sequencing, and *ICAM1* expression was measured using real-time PCR.

**Results.** Positive dominant model rs5498 participants had twice the risk of HCL (95% confidence interval (CI): [1.24–3.23], $P = 0.005$). The frequency of the G allele in rs5498 was 1.69 times higher in participants with HCL than in controls (95% CI [1.15–2.47], $P = 0.007$). Participants with the rs5498 AG or GG variants and high *ICAM1* mRNA expression ($\geq 3.12$) had 2.49 times the risk (95% CI [1.42–4.38], $P = 0.001$), and those with a high LDL-C concentration ($\geq 3.36$ mmol/L) had 2.09 times the risk (95% CI [1.19–3.66], $P = 0.010$) of developing ASCVD compared with those with low *ICAM1* mRNA and LDL-C levels. Interestingly, participants carrying the rs5498 AG or GG variants who had tachycardia (resting heart rates (RHRs) >100 beats/min) had a 5.02-times higher risk than those with a lower RHR (95% CI [1.35–18.63], $P = 0.016$).

**Conclusions.** It may consider the G allele in *ICAM1* rs5498 is associated with a higher risk of ASCVD in Thai people with HCL, and is also positively associated with *ICAM1* mRNA expression, LDL-C concentration, and RHR.

## INTRODUCTION

Atherosclerotic cardiovascular disease (ASCVD), which involves the accumulation of plaque in the subendothelial space of arterial walls, is a fundamental process in the development of various cardiovascular diseases, including coronary heart disease (CHD), cerebrovascular disease (CeVD), and peripheral artery disease (*Linton et al., 2019*; *Mach et al., 2019*). The many risk factors for ASCVD can be classified as modifiable risk factors, including dyslipidemia, high blood pressure, obesity, diabetes, alcohol consumption, and smoking, or non-modifiable risk factors, including age, sex, and genetic variation (*Arnett et al., 2019*; *Boehme, Esenwa & Elkind, 2017*; *Chen, Chang & Liou, 2020*; *Mach et al., 2019*; *Van Dijk et al., 2015*; *Virani et al., 2020*). Hypercholesterolemia (HCL) is a major risk factor for ASCVD that is a component of an abnormal lipid metabolism (*Martinez-Hervas & Ascaso, 2019*; *Verbeek et al., 2018*) and promotes endothelial dysfunction, which is associated with ischemic heart disease and stroke (*Grundy et al., 2004*; *Manktelow & Potter, 2009*; *O'Gara et al., 2013*).

Intercellular adhesion molecule 1 gene (*ICAM1*; CD54) is one of the immunoglobulin superfamily of cell adhesion molecules (CAMs) and is encoded by a gene located on chromosome 19p13.2 to 13.3. It plays a pivotal role in the firm attachment and transendothelial migration of leukocytes into the vascular intima. The *ICAM1* gene has been shown to be a potential biomarkers of endothelial dysfunction, which is the earliest stage in the pathogenesis of ASCVD (*Di Pietro, Formoso & Pandolfi, 2016*; *Herbert Haught et al., 1996*; *Sonja, Milica & Slađana, 2017*). *ICAM1* expression and function depends on its complement of single nucleotide polymorphisms (SNPs) and there are several known SNPs that are related to chronic inflammatory diseases, such as heart disease and stroke (*Gazi et al., 2014*; *Li, Qu & Dong, 2014*; *Shaker et al., 2010*; *Wang et al., 2015*). In Asian populations, rs5498 (A>G) and rs281432 (C>G) in the *ICAM1* gene have been reported to affect the risks of ASCVD and other cardiovascular diseases (CVDs) (*Chou et al., 2015*; *Yang et al., 2014*; *Yin et al., 2019*).

Resting heart rate (RHR) is a non-invasive clinical parameter that is regulated by the autonomic nervous system (ANS). The effect of inflammatory mediators on endothelial cells inhibits the effects of the ANS. The consequence is excessive sympathetic tone, which has marked deleterious effects on the vascular system, including an increase in heart rate, the induction of arrhythmias, such as tachycardia (RHR >100 beats/min) and atrial fibrillation (*Nasibullin et al., 2016*), and high blood pressure (*Böhm et al., 2015*; *Christofaro et al., 2017*; *Nanchen et al., 2013a*; *Nanchen et al., 2013b*; *Thayer, Yamamoto & Brosschot, 2010*). Furthermore, several previous studies have shown that a high heart rate affects several stages of the development of vascular diseases *via* endothelial function and genetic changes in several kinds of gene (*Custodis et al., 2010*; *Evans et al., 2019*; *Fox et al., 2007*; *Heusch, 2008*; *Palatini, 2007*; *Reil & Böhm, 2007*).

Although *ICAM1* gene variants have been reported to be important in patients with vascular diseases, there have been few studies of the relationships between *ICAM1* gene SNPs, RHR and HCL (*Li, Qu & Dong, 2014*; *Nasibullin et al., 2016*; *Nepal, Yadav & Kong, 2019*). Therefore, we conducted a cross-sectional study to explore the relationships between

the *ICAM1* SNPs, rs5498 (A>G) and rs281432 (C>G), HCL, and ASCVD in Thai adults. We hypothesized that individuals with HCL who carry these genetic variants would show higher gene expression, impaired ANS function, and have a greater risk of ASCVD. If so, the identification of these polymorphisms would be useful for the prediction and prevention of ASCVD.

## MATERIALS AND METHODS

### Ethics statement
The study protocol was approved by the ethics committee of the Faculty of Tropical Medicine, Mahidol University (TMEC 18-026), and written informed consent was obtained from all the participants.

### Study design and participants
We conducted a cross-sectional study in which we randomly recruited 278 adults who had no evidence of systemic inflammation or infectious diseases in Sung Noen District, Nakhon Ratchasima Province, Thailand. The participants comprised 85 men and 193 women, aged between 35 and 60 years. The exclusion criteria were: (1) pregnancy or lactation; (2) presence of a serious health condition, including ischemic heart disease, stroke, peripheral artery disease, or any chronic disease; and (3) the regular use of medication. The participants were classified into two groups according to their lipid profile: 143 controls and 135 with HCL. The presence of HCL was diagnosed according to the National Cholesterol Education Program Adult Treatment Panel III (NCEP ATP III), as follows: serum total cholesterol (TC) $\geq$ 5.17 mmol/L, triglycerides (TG) $\geq$ 1.70 mmol/L, low-density lipoprotein-cholesterol (LDL-C) $\geq$ 3.37 mmol/L, high-density lipoprotein-cholesterol (HDL-C) $\leq$ 1.04 mmol/L for men and $\leq$ 1.29 mmol/L in women, and TG/HDL-C ratio $\geq$ 1.75 mmol/L. Each of these cut-off values is borderline high according to the reference ranges (*Expert Panel on Detection, Evaluation, and Treatment of High Blood Cholesterol in Adults, 2001*).

### Clinical and laboratory evaluation
The participants underwent a physical examination to determine their health status. RHR and blood pressure were measured after 15 min of rest in the sitting position using an automatic oscillometer (Riester, Jungingen, Germany). Anthropometric measurements were made and classified according to the World Health Organization (WHO) Asia-Pacific guidelines (*WHO, 2000*; *WHO, 2004*). Body mass index (BMI) was calculated as body mass divided by height squared (kg/m$^2$). Waist and hip circumference were measured using an inelastic tape-measure and waist-to-hip ratio (WHR) was calculated. These procedures were performed by the same trained nurses and researchers, using standardized techniques. Blood samples were collected from peripheral blood vessels after at least 8 h of fasting to determine lipid profile. Serum TC, TG, and HDL-C concentrations were determined using enzymatic colorimetric methods on a Cobas 6000 analyzer (Roche Diagnostics International Ltd., Basel, Switzerland). Serum low-density lipoprotein-cholesterol (LDL-C) concentration was calculated using the Friedewald equation in participants with a

TG concentration <4.5 mmol/L as follows: [LDL-C] = [TC] − ([TG]/2.2) − [HDL-C] (*Friedewald, Levy & Fredrickson, 1972*).

## Genomic DNA extraction, genotyping of the *ICAM1* rs5498 (AG) and rs281432 (CG) polymorphisms, and DNA sequencing

Genomic DNA was extracted from peripheral blood leukocytes using a FlexiGene DNA kit (Qiagen, Hilden, Germany), according to the manufacturer's instructions. The *ICAM1* rs5498 and rs281432 polymorphisms were identified by PCR using KAPA2G Fast HotStart master mix (Kapa Biosystems, Wilmington, MA, USA) containing 1 µL of template DNA and 0.5 µL of each primer (forward and reverse). The final volume was adjusted to 25 µL with nuclease-free water. The PCR amplification conditions comprised an initial denaturation step of 5 min at 95 °C, followed by 35 cycles of denaturation for 30 s at 95 °C, annealing for 45 s at 60 °C, and extension for 30 s at 72 °C, with a final extension of 5 min at 72 °C. The primers used for *ICAM1* rs5498 were (forward) 5′-TTG AGG GCA CCT ACC TCT GT-3′ and (reverse) 5′-CAT TAT GAC TGC GGC TGC TA-3′, yielding an amplicon of size 214 bp, and those for rs281432 were (forward) 5′-GAG GAG CTG GGA CTT CCC TT-3′ and (reverse) 5′-CCC TGA CCT GCA GTC CTT TA-3′, yielding an amplicon of 231 bp. The amplified DNA was evaluated by agarose gel electrophoresis. The *ICAM1* rs5498 and rs281432 regions were also directly genotyped by DNA sequencing by Bio Basic Asia Pacific Pte. Ltd. (Bukit Batok, Singapore).

## RNA extraction and real-time PCR analysis

To measure *ICAM1* mRNA expression in peripheral blood mononuclear cells, RNA was isolated using NucleoZol reagent (Macherey-Nagel, Duren, Germany), according to the manufacturer's instructions. To synthesize cDNA for real-time PCR, 2 µg RNA was reverse transcribed and genomic DNA was removed using a ReverTra Ace® qPCR RT Master Mix and gDNA remover from Toyobo Co. Ltd. (Osaka, Japan). Real-time PCR was performed according to the manufacturer's instructions using iTaq™ Universal SYBR® Green Supermix (Bio-Rad Laboratories Inc., Hercules, CA, USA). The primer sequences were as follows: (forward) 5′-GCA TCC TGG GCT ACA CTG AG-3′ and (reverse) 5′-TGC TGT AGC CAA ATT CGT TG-3′ for the reference gene *GAPDH*, and (forward) 5′-ACA GTC ACC TAT GGC AAC GAC-3′ and (reverse) 5′-GTC ACT GTC TGC AGT GTC CT-3′ for *ICAM1*. The relative *ICAM1* mRNA expression was normalized to that of *GAPDH*. All amplification reactions were performed in duplicate and the results were quantified using the $2^{-\Delta\Delta Ct}$ method.

## Statistical analysis

Statistical analyses were conducted using SPSS 18.0 (IBM, Inc., Armonk, NY US). The continuous data were first tested for normal distribution using the Kolmogorov–Smirnov test. If the data were normally distributed, differences of mean between two groups were identified using Student's *t* test. One-way analysis of variance (ANOVA) was used to compare data more than two groups and followed by a *post-hoc* analysis. For multiple testing, the results for *ICAM1* rs5498 genotype, including reference allele and variant genotype among participants were corrected using a Bonferroni correction, based on

the number of calculated ($n = 6$). This study set significance at a Bonferroni corrected alpha ($\alpha$) [corrected $P < 0.008$, $\alpha = 0.050/6$]. Meanwhile, the Mann–Whitney $U$ test or Kruskal-Wallis test for non-normally distributed data. Twin sets of categorical data were compared using the chi-square test. The genotype frequencies of the *ICAM1* variants were analyzed for Hardy-Weinberg equilibrium (HWE) in the HCL and control groups. To determine the relationships between *ICAM1* SNPs and HCL, odds ratios (ORs) and 95% confidence intervals (95% CIs) were calculated using logistic regression. Comparisons of the distributions of the alleles and genotypes were performed using the chi-square test. Statistical significance was set at $P < 0.05$.

## RESULTS

### Clinical and laboratory characteristics of the participants with HCL and controls

Table 1 shows the clinical and laboratory characteristics of participants with HCL and controls. The prevalence of HCL was 49% (135/278). The sex and age ratios in the groups were identical. The SBP, DBP, and RHR of the participants were significantly higher than those of controls. Likewise, the body composition of the groups differed: BMI, WC, HC, and WHR were significantly higher in the HCL group than the control group. As expected, the serum TC, LDL-C, and TG concentrations and the TG/HDL-C ratio of the HCL group were significantly higher than those of controls (all $P < 0.001$). Moreover, there were no significant differences in behavioral characteristics, including in the prevalences of cigarette smoking, alcohol consumption, and exercise, between the HCL and control groups. However, there were no differences in HDL-C concentration or *ICAM1* mRNA expression between the two groups ($P = 0.218$ and $P = 0.096$, respectively).

### *ICAM1* rs5498 and rs281432 genotype

The rs5498 and rs281432 polymorphisms of *ICAM1* were genotyped. As shown in Fig. 1, the heterozygous and homozygous variants of *ICAM1* rs5498 were identified as AG and GG, while AA genotype was the reference allele, and those of rs281432 were CG and GG, and CC, respectively.

### Associations of the *ICAM1* rs5498 and rs281432 polymorphisms with HCL

The distributions of the genotypes were consistent with the presence of Hardy-Weinberg equilibrium in both groups. As shown in Table 2, the genotype distribution of *ICAM1* rs5498 (A > G) in the participants with HCL was GG (10%), AG (43%), and AA (47%), which was significantly different to that of the control participants, who had a distribution of GG (7.0%), AG (29%), and AA (64%). Allele frequency analysis for *ICAM1* rs5498 showed that the G allele was more frequent in the participants with HCL than in controls (OR = 1.69, 95% CI [1.15–2.47], $P = 0.007$). Moreover, the genetic model of *ICAM1* rs5498 was evaluated, which demonstrated a statistically significant association between *ICAM1* rs5498 and HCL in the dominant model (AA vs. AG+GG) (OR = 2.00, 95% CI [1.24–3.23], $P = 0.005$). This also showed a positive association between the presence of

**Table 1** Clinical and laboratory characteristics of the participants with HCL and controls.

| Variables | Control | HCL | *P* value |
|---|---|---|---|
| *N* | 143 | 135 | – |
| Age, (years) | 46.56 ± 6.47 | 48.06 ± 6.00 | 0.065 |
| Male (%) | 32.9 | 28.1 | 0.393 |
| Smoker (%) | 21.7 | 25.9 | 0.405 |
| Current alcohol consumer (%) | 37.8 | 40.7 | 0.611 |
| Exercise (%) | 9.1 | 8.9 | 0.953 |
| SBP (mmHg) | 122.42 ± 14.70 | 128.48 ± 15.30 | **0.001** |
| DBP (mmHg) | 76.59 ± 10.05 | 80.13 ± 9.87 | **0.003** |
| RHR (beats/min) | 77.44 ± 12.13 | 81.04 ± 11.53 | **0.012** |
| BMI (kg/m$^2$) | 25.70 ± 5.01 | 27.30 ± 4.62 | **0.006** |
| WC (cm) | 85.70 ± 12.31 | 89.27 ± 10.34 | **0.009** |
| HC (cm) | 95.59 ± 10.31 | 97.87 ± 8.89 | **0.049** |
| WHR | 0.89 ± 0.06 | 0.91 ± 0.06 | **0.023** |
| TC (mmol/L) | 4.30 ± 0.65 | 5.47 ± 1.01 | **<0.001** |
| LDL-C (mmol/L) | 2.48 ± 0.53 | 3.33 ± 0.90 | **<0.001** |
| HDL-C (mmol/L) | 1.30 ± 0.32 | 1.25 ± 0.34 | 0.218 |
| TG (mmol/L) | 1.10 (0.41, 2.94) | 1.90 (0.60, 4.46) | **<0.001**[*] |
| TG/HDL-C ratio | 0.89 (0.22, 2.44) | 1.49 (0.34, 8.62) | **<0.001**[*] |
| *ICAM1* mRNA expression (arbitrary units) | 1.15 (0.14, 9.19) | 1.51 (0.15, 9.97) | 0.096[*] |

**Notes.**

HCL, hypercholesterolemia; SBP, systolic blood pressure; DBP, diastolic blood pressure; RHR, resting heart rate; BMI, body mass index; WC, waist circumference; HC, hip circumference; WHR, waist to hip ratio; TC, total cholesterol; LDL-C, low-density lipoprotein-cholesterol; HDL-C, high-density lipoprotein-cholesterol; TG, triglycerides; TG/HDL-C ratio, triglyceride-to-high-density lipoprotein-cholesterol ratio; *ICAM1*, Intercellular Adhesion Molecule 1.

*P* values calculated using the $\chi^2$ test (categorical variables) or Student's *t* test (continuous variables with normal distribution, the data represent as mean ± S.D).

[*]*P* values calculated using the Mann–Whitney *U* test (continuous variables with non-normal distribution, the data represent as median (min, max). *P* value < 0.05 was considered statistically significant.

A smoker was defined who had smoked at least 100 cigarettes in their lifetime. Moderate and heavy alcohol drinking were classified as a current alcohol consumer. Exercise was defined as activity of a moderate or vigorous intensity for ≥30 min/day on at least 3 days a week.

*P* value < 0.05 (in bold) was considered statistically significant.

both HCL and *ICAM1* rs5498 and a high risk of ASCVD. However, there was no significant difference in the genotype distribution of *ICAM1* rs281432 (C >G) between the HCL and control groups. The risk factors in genetic models of rs281432 (codominant, dominant, and recessive) were compared by logistic regression analysis and no significant associations were found using these genetic models.

## Comparison of the clinical and laboratory characteristics of participants with rs5498 and rs281432 of *ICAM1* polymorphisms using the dominant model

We found strong associations between the SNPs and HCL using the dominant model (Table 2). Therefore, we further analyzed the relationships between *ICAM1* SNPs, HCL, and other variables. The mean values of clinical and laboratory characteristics were calculated according to allele distribution (Table 3). Participants carrying the G allele (AG and GG genotype) of rs5498 had a higher RHR than those carrying only the A allele

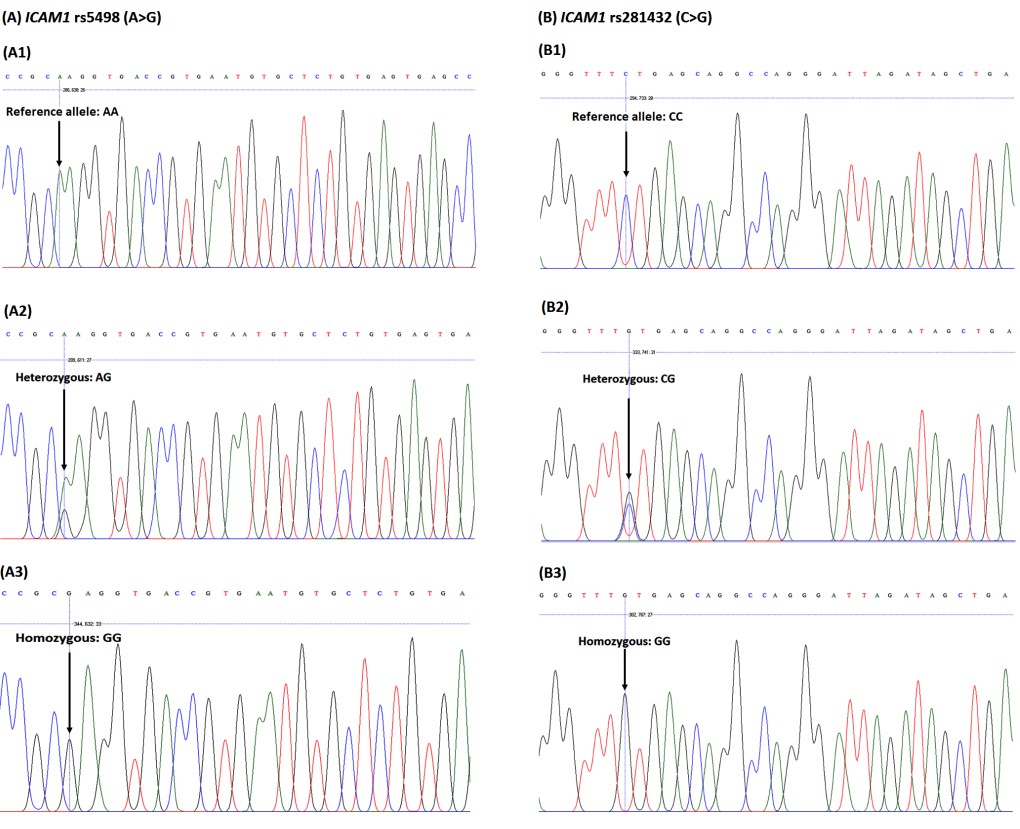

**Figure 1** **DNA sequencing chromatograph for the *ICAM1* (A) rs5498 (A > G) polymorphism and (B) rs281432 (C >G) polymorphism.** The *ICAM1* rs5498 (A1) reference allele, homozygous genotype: AA; (A2) heterozygous genotype: AG; (A3) homozygous genotype: GG. The *ICAM1* rs281432 (B1) reference allele, homozygous genotype: CC; (B2) heterozygous genotype: CG); (B3) homozygous genotype: GG.

(AA *vs.* AG + GG: 77.26 ± 11.44 bpm *vs.* 81.58 ± 12.19 bpm; $P = 0.003$). Furthermore, participants carrying the G allele of rs5498 had higher serum TC and LDL-C concentrations and *ICAM1* mRNA expression than those carrying only the A allele ($P = 0.004$, 0.002, and <0.001, respectively). Moreover, the prevalence of smoking was higher in rs5498 reference allele (AA) participants than in those with the variants (AG+GG) ($P = 0.035$). However, there were no differences between these groups with respect to alcohol consumption, exercise, serum TG concentration, or TG/HDL-C ratio. Furthermore, there were no significant relationships between the *ICAM1* rs281432 genotype and clinical and laboratory characteristics.

## Relationships between *ICAM1* rs5498 polymorphism, HCL, and other ASCVD risk factors

According to the NCEP ATP III and previous studies, there are multiple ASCVD risk factors related to HCL, including overweight/obesity, dyslipidemia, high *ICAM1* mRNA expression, and *ICAM1* SNPs. Therefore, we assessed the ASCVD risk factors according to the genetic dominant model in participants with HCL with the G allele (variants, AG+GG) and in those without the G allele (reference allele, AA) (Fig. 2). We found that
**Table 2  Genetic models analysis for the *ICAM1* rs5498 and rs281432 polymorphisms in HCL and control participants.**

| Variables | All participants (*n* = 278) | Control (*n* = 143) | HCL (*n* = 135) | OR (95% CI) | *P* value |
|---|---|---|---|---|---|
| *ICAM1* rs5498 allele, *n* (%) | | | | | |
| A | 408 (73.0) | 224 (78.0) | 184 (68.0) | (Reference) | |
| G | 148 (27.0) | 62 (22.0) | 86 (32.0) | 1.69 (1.15–2.47) | **0.007** |
| $P_{HWE}$ | 0.220 | 1.000 | 0.140 | | |
| *ICAM1* rs5498 genotypes | | | | | |
| Codominant effects, *n* (%) | | | | | |
| AA | 154 (55.0) | 91 (64.0) | 63 (47.0) | (Reference) | |
| AG | 100 (36.0) | 42 (29.0) | 58 (43.0) | 1.99 (1.20–3.32) | **0.008** |
| GG | 24 (9.0) | 10 (7.0) | 14 (10.0) | 2.02 (0.84–4.84) | 0.114 |
| Dominant effects, *n* (%) | | | | | |
| AA | 154 (55.0) | 91 (64.0) | 63 (47.0) | (Reference) | |
| AG + GG | 124 (45.0) | 52 (36.0) | 72 (53.0) | 2.00 (1.24–3.23) | **0.005** |
| Recessive effects, *n* (%) | | | | | |
| AA + AG | 254 (91.0) | 133 (93.0) | 121 (90.0) | (Reference) | |
| GG | 24 (9.0) | 10 (7.0) | 14 (10.0) | 1.54 (0.66–3.59) | 0.319 |
| *ICAM1* rs281432 allele, *n* (%) | | | | | |
| C | 362 (65.0) | 185 (65.0) | 177 (66.0) | (Reference) | |
| G | 194 (35.0) | 101 (35.0) | 93 (34.0) | 0.96 (0.68–1.36) | 0.859 |
| $P_{HWE}$ | 0.360 | 0.580 | 0.570 | | |
| *ICAM1* rs281432 genotypes | | | | | |
| Codominant effects, *n* (%) | | | | | |
| CC | 114 (41.0) | 58 (41.0) | 56 (41.0) | (Reference) | |
| CG | 134 (48.2) | 69 (48.0) | 65 (48.0) | 0.98 (0.59–1.61) | 0.923 |
| GG | 30 (10.8) | 16 (11.0) | 14 (10.0) | 0.91 (0.40–2.03) | 0.811 |
| Dominant effects, *n* (%) | | | | | |
| CC | 114 (41.0) | 58 (41.0) | 56 (41.0) | (Reference) | |
| CG +GG | 164 (59.0) | 85 (59.0) | 79 (58.0) | 0.96 (0.60–1.55) | 0.876 |
| Recessive effects, *n* (%) | | | | | |
| CC +CG | 248 (89.2) | 127 (89.0) | 121 (89.0) | (Reference) | |
| GG | 30 (10.8) | 16 (11.0) | 14 (10.0) | 0.92 (0.43–1.96) | 0.826 |

**Notes.**

*P* values assessed using odds ratios, according to the genetic model. $P_{HWE}$ for the Hardy-Weinberg equilibrium test. Genetic models: codominant model (reference allele *vs.* heterozygous variants and reference allele *vs.* homozygous variants), dominant model (reference allele *vs.* heterozygous variants + homozygous variants), and recessive model (reference allele + heterozygous variants *vs.* homozygous variants). *P* value < 0.05 was considered statistically significant.

the mean RHR of participants with HCL and the AG or GG genotype was significantly higher than that of controls with the AA genotype ($P = 0.002$). In addition, the BMI of participants with HCL and controls with the AG or GG genotype was higher than that of participants with HCL and controls with the AA genotype ($P = 0.042$ for trend). There were also significant differences in lipid profile: participants with the AG or GG genotypes had significantly higher TC, LDL-C, and TG/HDL ratio than in controls with any of the genotypes ($P < 0.001$). The TG concentration in participants with AG or GG was higher than in controls with any genotype or in participants with HCL and the AA genotype

**Table 3  Clinical and laboratory characteristics of participants with rs5498 or rs281432 ICAM1 variants in the dominant model.**

| Variables | ICAM1 rs5498 | | P value | ICAM1 rs281432 | | P value |
|---|---|---|---|---|---|---|
| | AA (n = 154) | AG + GG (n = 124) | | CC (n = 114) | CG + GG (n = 164) | |
| Age (years) | 47.66 ± 6.40 | 46.81 ± 6.13 | 0.264 | 47.65 ± 5.92 | 47.04 ± 6.53 | 0.428 |
| Male (%) | 34.4 | 25.8 | 0.121 | 28.1 | 32.3 | 0.450 |
| Smoker (%) | 28.6 | 17.7 | **0.035** | 24.6 | 23.2 | 0.789 |
| Current alcohol consumer (%) | 40.3 | 37.9 | 0.689 | 41.2 | 37.8 | 0.565 |
| Exercise (%) | 9.7 | 8.1 | 0.627 | 9.6 | 8.5 | 0.750 |
| SBP (mmHg) | 125.66 ± 15.73 | 124.99 ± 14.73 | 0.717 | 127.25 ± 17.34 | 124.05 ± 13.55 | 0.085 |
| DBP (mmHg) | 78.38 ± 9.71 | 78.22 ± 10.61 | 0.892 | 79.17 ± 11.29 | 77.71 ± 9.18 | 0.257 |
| RHR (beats/min) | 77.26 ± 11.44 | 81.58 ± 12.19 | **0.003** | 79.19 ± 12.18 | 79.18 ± 11.83 | 0.995 |
| BMI (kg/m$^2$) | 26.17 ± 4.88 | 26.87 ± 4.87 | 0.235 | 26.21 ± 5.16 | 26.66 ± 4.69 | 0.448 |
| WC (cm) | 86.75 ± 11.78 | 88.28 ± 11.16 | 0.272 | 87.19 ± 11.96 | 87.60 ± 11.23 | 0.773 |
| HC (cm) | 95.94 ± 9.83 | 97.65 ± 9.48 | 0.143 | 96.19 ± 10.21 | 97.05 ± 9.33 | 0.466 |
| WHR | 0.90 ± 0.06 | 0.90 ± 0.06 | 0.972 | 0.91 ± 0.06 | 0.90 ± 0.06 | 0.607 |
| TC (mmol/L) | 4.70 ± 1.00 | 5.06 ± 1.03 | **0.004** | 4.88 ± 1.13 | 4.85 ± 0.95 | 0.812 |
| LDL-C (mmol/L) | 2.75 ± 0.81 | 3.06 ± 0.86 | **0.002** | 2.87 ± 0.89 | 2.90 ± 0.82 | 0.724 |
| HDL-C (mmol/L) | 1.27 ± 0.34 | 1.28 ± 0.33 | 0.824 | 1.28 ± 0.35 | 1.26 ± 0.32 | 0.609 |
| TG (mmol/L) | 1.33 (0.41, 4.46) | 1.33 (0.47, 3.99) | 0.740[*] | 1.40 (0.41, 4.46) | 1.29 (0.44, 3.53) | 0.608[*] |
| TG/HDL-C ratio | 1.03 (0.29, 8.62) | 1.07 (0.22, 5.31) | 0.939[*] | 1.07 (0.30, 8.62) | 1.05 (0.22, 6.90) | 0.917[*] |
| ICAM1 mRNA expression | 1.05 (0.15, 9.97) | 1.84 (0.14, 9.94) | **<0.001**[*] | 1.14 (0.14, 9.97) | 1.46 (0, 9.92) | 0.302[*] |

Notes.
P values calculated using Student's t test (continuous variables with normal distribution, the data represent as mean ± S.D.).
*P values calculated using the Mann–Whitney U test (continuous variables with non-normal distribution, the data represent as median (min, max). P value <0.05 was considered statistically significant. Dominant model analysis: reference allele (AA) vs. heterozygous + homozygous variants (AG + GG).

($P < 0.001$). Considering, individual HCL group, the TG concentration in participants with AG or GG was higher than in HCL participants with an AA genotype ($P = 0.004$). Furthermore, we found higher expression of ICAM1 mRNA in HCL participants with AG or GG genotypes than in control group with an AA genotype ($P = 0.013$). However, there were no differences in HDL-C concentrations among the groups. These results imply that individuals carrying the G allele of rs5498, whether hypercholesterolemic or normocholesterolemic, have a higher risk of ASCVD. In contrast, participants carrying the rs281432 variants did not show this association.

## The effect of the interaction of the ICAM1 rs5498 polymorphism and ASCVD risk factors

The combined impacts of ICAM1 genotype and specific ASCVD risk factors were determined. In the participants, values above the 75th percentile for RHR (≥86 beats/min) or tachycardia (>100 beats/min), abnormal lipid profile (borderline high values for each parameter), and values above the 75th percentile for ICAM1 mRNA expression (≥3.12 arbitrary units) were defined as confering a high risk of ASCVD. Participants with the rs5498 AG+GG genotype and a high serum LDL-C concentration (≥3.36 mmol/L) were at higher risk of ASCVD (OR = 2.09, 95% CI [1.19–3.66], $P = 0.010$) than those with a low serum LDL-C concentration (Table 4). There was also a 2.49-fold higher ASCVD risk (95%

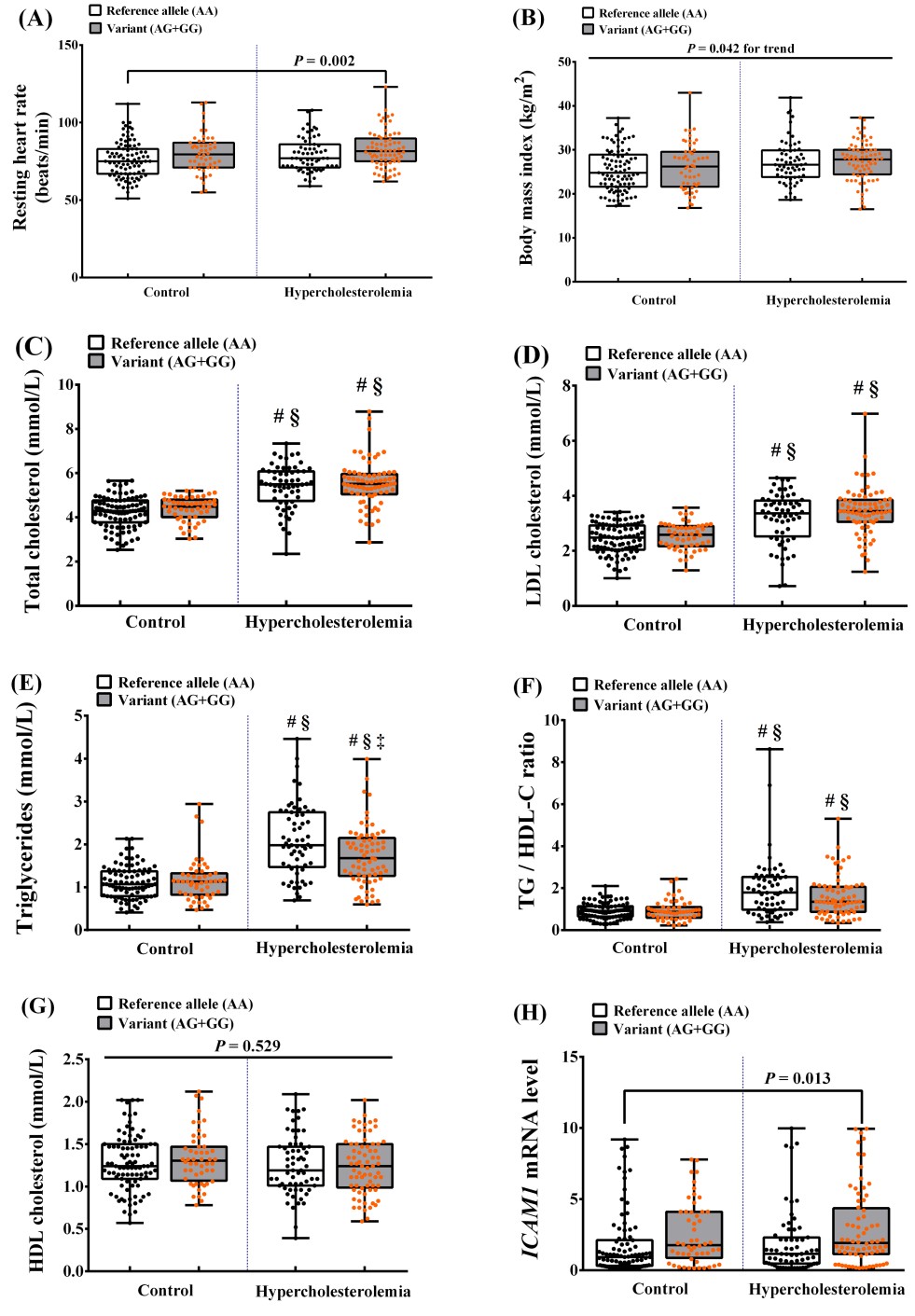

**Figure 2** **Relationships between the *ICAM1* rs5498 polymorphism genotype in the dominant model and ASCVD risk factors.** Control (reference allele AA, $n = 91$; variant AG + GG, $n = 52$) and HCL (reference allele AA, $n = 63$; variant AG + GG, $n = 72$) groups. *P* values were calculated using a one-way analysis of variance (ANOVA), followed by a *post-hoc* analysis to detect statistical difference among groups for normal distribution data. (continued on next page…)

**Figure 2 (...continued)**
Meanwhile, *P* values for non-normal distribution data were tested using *Kruskal-Wallis test*, followed by Mann–Whitney *U* test to detect statistical difference among groups. (A) Resting heart rate (RHR), $P = 0.002$ *vs.* control (AA). (B) Body mass index (BMI), $P = 0.042$ for trend. (C) Total cholesterol (TC), #$P < 0.001$ *vs.* control (AA), §$P < 0.001$ *vs.* control (AG + GG). (D) LDL cholesterol (LDL-C), #$P < 0.001$ *vs.* control (AA), §$P < 0.001$ *vs.* control (AG + GG). (E) Triglycerides (TG), #$P < 0.001$ *vs.* control (AA), §$P < 0.001$ *vs.* control (AG + GG), †$P = 0.004$ *vs.* HCL (AA). (F) TG/HDL-C ratio, #$P < 0.001$ *vs.* control (AA), §$P < 0.001$ *vs.* control (AG + GG). (G) HDL cholesterol (HDL-C), $P = 0.529$. (H) *ICAM1* mRNA expression, $P = 0.013$ *vs.* control (AA).

CI [1.42–4.38], $P = 0.001$) in participants with an *ICAM1* mRNA expression level of ≥ 3.12 than in those with an *ICAM1* mRNA expression level of <3.12 (Table 5). Furthermore, we found a robust interaction between the *ICAM1* rs5498 polymorphism and high RHR. The presence of the rs5498 variant (AG+GG) and RHR above the 75th percentile or tachycardia increased the trend risk of ASCVD (OR = 1.58, 95% CI [0.91–2.74], $P = 0.101$, and OR = 5.02, 95% CI [1.35–18.63], $P = 0.016$, respectively) over that associated with a lower RHR (Table 4). This finding suggests that the higher the RHR, the higher the risk of developing ASCVD. All the ORs quoted were adjusted for age, gender, BMI, smoking status, alcohol consumption status, and exercise habits (Tables 4 and 5). However, we did not identify any significant interactions between the *ICAM1* rs281432 polymorphism and ASCVD risk factors (Table S1).

## DISCUSSION

In the present cross-sectional study we investigated the relationships between the *ICAM1* rs5498 polymorphism and modifiable and non-modifiable risk factors for ASCVD. First, we found a significant association between the rs5498 dominant model and HCL in a sample of adult Thai people. Second, rs5498 variants were associated with multiple risk factors for ASCVD: RHR, TC, LDL-C, and *ICAM1* mRNA expression. Third, participants with HCL and an rs5498 variant had higher RHR, BMI, TC, LDL-C, TG, TG/HDL-C ratio, and *ICAM1* mRNA expression. Finally, participants with an rs5498 variant had a higher RHR, higher serum LDL-C concentration, and higher expression of *ICAM1* mRNA than those with the reference allele AA genotype. These results indicate that in Thai adults variation at the *ICAM1* rs5498 locus may associated with HCL and high RHR, which are risk factors for ASCVD, and this is consistent with the findings of previous studies conducted in other populations (*Iwao, Morisaki & Morisaki, 2004*; *Sarecka-Hujar, Zak & Krauze, 2009*).

The pathogenesis of ASCVD is considered to be multifactorial, involving oxidative stress and an upregulation of *CAMs* expression, induced by proinflammatory cytokines, including interleukin (IL)-6, tumor necrosis factor (TNF)- $\alpha$, and IL-1 $\beta$, all of which are consequences of overweight/obesity, HCL, and related risk factors, including smoking and alcohol consumption (*Boehme, Esenwa & Elkind, 2017*; *Nepal, Yadav & Kong, 2019*). The accumulation of lipids, especially LDL-C, is thought to initiate oxidative stress, and oxidized LDL and macrophage recruitment amplify the inflammation, resulting in endothelial dysfunction in arterial walls (*Klop, Elte & Castro Cabezas, 2013*; (*Linton et al., 2019*). High expression of *ICAM1* is a result of the chronic inflammation of

**Table 4  The effect of the interaction of the *ICAM1* rs5498 polymorphism and ASCVD risk factors.**

| Variables | ASCVD risk factors | | *P* value | OR[a] (95% CI) | *P*[a] value | OR[b] (95% CI) | *P*[b] value |
|---|---|---|---|---|---|---|---|
| | Low-risk | High-risk | | | | | |
| 75th percentile RHR (beats/min), *n* (%) | <86 (*n* = 203) | ≥ 86 (*n* = 75) | | | | | |
| *ICAM1* rs5498 (A >G) | | | | | | | |
| AA | 120 (59.1) | 34 (45.3) | | (Reference) | | (Reference) | |
| AG + GG | 83 (40.9) | 41 (54.7) | **0.040** | 1.74 (1.02–2.97) | **0.041** | 1.58 (0.91–2.74) | 0.101 |
| Tachycardia at RHR (beats/min), *n* (%) | ≤ 100 (*n* = 263) | >100 (*n* = 15) | | | | | |
| *ICAM1* rs5498 (A >G) | | | | | | | |
| AA | 151 (57.4) | 3 (20.0) | | (Reference) | | (Reference) | |
| AG + GG | 112 (42.6) | 12 (80.0) | **0.005** | 5.40 (1.49–19.56) | **0.010** | 5.02 (1.35–18.63) | **0.016** |
| BP (mmHg), *n* (%) | ≤ 130 and/or 85 (*n* = 221) | >130 and/or 85 (*n* = 57) | | | | | |
| *ICAM1* rs5498 (A >G) | | | | | | | |
| AA | 126 (57.0) | 28 (49.1) | | (Reference) | | (Reference) | |
| AG + GG | 95 (43.0) | 29 (50.9) | 0.285 | 1.37 (0.77–2.46) | 0.286 | 1.31 (0.71–2.41) | 0.390 |
| TC (mmol/L), *n* (%) | <5.17 (*n* = 183) | ≥ 5.17 (*n* = 95) | | | | | |
| *ICAM1* rs5498 (A >G) | | | | | | | |
| AA | 108 (59.0) | 46 (48.4) | | (Reference) | | (Reference) | |
| AG + GG | 75 (41.0) | 49 (51.6) | 0.092 | 1.59 (0.96–2.63) | 0.071 | 1.60 (0.96–2.69) | 0.073 |
| LDL-C (mmol/L), *n* (%) | <3.36 (*n* = 205) | ≥ 3.36 (*n* = 73) | | | | | |
| *ICAM1* rs5498 (A >G) | | | | | | | |
| AA | 122 (59.5) | 32 (43.8) | | (Reference) | | (Reference) | |
| AG + GG | 83 (40.5) | 41 (56.2) | **0.021** | 1.99 (1.15–3.44) | **0.014** | 2.09 (1.19–3.66) | **0.010** |
| HDL-C (mmol/L), *n* (%) | >1.29 (women) >1.04 (men) (*n* = 149) | ≤ 1.29 (women) ≤ 1.04 (men) (*n* = 129) | | | | | |
| *ICAM1* rs5498 (A >G) | | | | | | | |
| AA | 83 (55.7) | 71 (55.0) | | (Reference) | | (Reference) | |
| AG + GG | 66 (44.3) | 58 (45.0) | 0.911 | 1.03 (0.64–1.65) | 0.911 | 1.04 (0.63–1.70) | 0.886 |
| TG (mmol/L), *n* (%) | <1.69 (*n* = 191) | ≥ 1.69 (*n* = 87) | | | | | |
| *ICAM1* rs5498 (A >G) | | | | | | | |
| AA | 106 (55.5) | 48 (55.2) | | (Reference) | | (Reference) | |
| AG + GG | 85 (44.5) | 39 (44.8) | 0.960 | 1.01 (0.61–1.69) | 0.960 | 1.02 (0.59–1.75) | 0.943 |

**Table 4** (*continued*)

| Variables | ASCVD risk factors | | P value | ORᵃ (95% CI) | Pᵃ value | ORᵇ (95% CI) | Pᵇ value |
|---|---|---|---|---|---|---|---|
| | Low-risk | High-risk | | | | | |
| TG/HDL-C (mmol/L), *n* (%) | <1.75 (*n* = 214) | ≥ 1.75 (*n* = 64) | | | | | |
| *ICAM1* rs5498 (A >G) | | | | | | | |
| AA | 119 (55.6) | 35 (54.7) | | (Reference) | | (Reference) | |
| AG + GG | 95 (44.4) | 29 (45.3) | 0.897 | 1.04 (0.59–1.82) | 0.897 | 0.98 (0.55–1.75) | 0.936 |

Notes.

ASCVD, Atherosclerotic cardiovascular disease.

*P* value calculated using the $\chi^2$ test. ORᵃ Unadjusted. ORᵇ Adjusted for age, gender, BMI, smoking status, alcohol consumption status, and exercise habits. *Pᵃ* value associated with odds ratio (OR)ᵃ. *Pᵇ* value associated with ORᵇ. *P* value < 0.05 was considered statistically significant.

**Table 5** The interaction of the *ICAM1* rs5498 and rs281432 polymorphism with mRNA expression.

| Variables | ASCVD risk factors | | P value | ORᵃ (95% CI) | Pᵃ value | ORᵇ (95% CI) | Pᵇ value |
|---|---|---|---|---|---|---|---|
| | Low-risk | High-risk | | | | | |
| 75th percentile (arbitrary units) *ICAM1* mRNA expression, *n* (%) | <3.12 (*n* = 206) | ≥ 3.12 (*n* = 72) | | | | | |
| *ICAM1* rs5498 (A >G) | | | | | | | |
| AA | 127 (61.7) | 27 (37.5) | | (Reference) | | (Reference) | |
| AG + GG | 79 (38.3) | 45 (62.5) | <0.001 | 2.68 (1.54–4.66) | <0.001 | 2.49 (1.42–4.38) | **0.001** |
| *ICAM1* rs281432 (C >G) | | | | | | | |
| CC | 88 (42.7) | 26 (36.1) | | (Reference) | | (Reference) | |
| CG +GG | 118 (57.3) | 46 (63.9) | 0.326 | 1.32 (0.76–2.30) | 0.327 | 1.38 (0.78–2.44) | 0.266 |

Notes.

ASCVD, Atherosclerotic cardiovascular disease.

*P* value calculated using the $\chi^2$ test. ORᵃ Unadjusted. ORᵇ Adjusted for age, gender, BMI, smoking status, alcohol consumption status, and exercise habits. *Pᵃ* value associated with odds ratio (OR)ᵃ. *Pᵇ* value associated with ORᵇ. *P* value <0.05 was considered statistically significant.

the arteries and leads to the migration of leukocytes into the intima (*Herbert Haught et al., 1996*). The common polymorphism at rs5498 in exon 6 of the *ICAM1* gene results in the substitution of glutamate for lysine (K469E) in the immunoglobulin-like domain 5 of the ICAM-1 protein, which is the result of an A-to-G substitution (*Liu, Wu & Liu, 2013*). This polymorphism has also been suggested to affect mRNA splicing patterns, with effects on cell–cell interactions and the inflammatory response (Fig. 3) (*Iwao, Morisaki & Morisaki, 2004*; *Sarecka-Hujar, Zak & Krauze, 2009*). These changes have been shown to be involved in the etiology of ASCVD (*Gaetani et al., 2002*).

A previous study by *Sarecka-Hujar, Zak & Krauze (2009)* showed that individuals possessing the AG or GG genotypes at *ICAM1* rs5498 tended to have higher TC and LDL-C concentrations than those possessing the AA genotype, and were therefore at higher risk of developing coronary artery disease. This is consistent with the present finding of significant positive relationships between the rs5498 variant and high serum TC and LDL-C concentrations. In addition, we identified associations with other risk factors (BMI, TG,

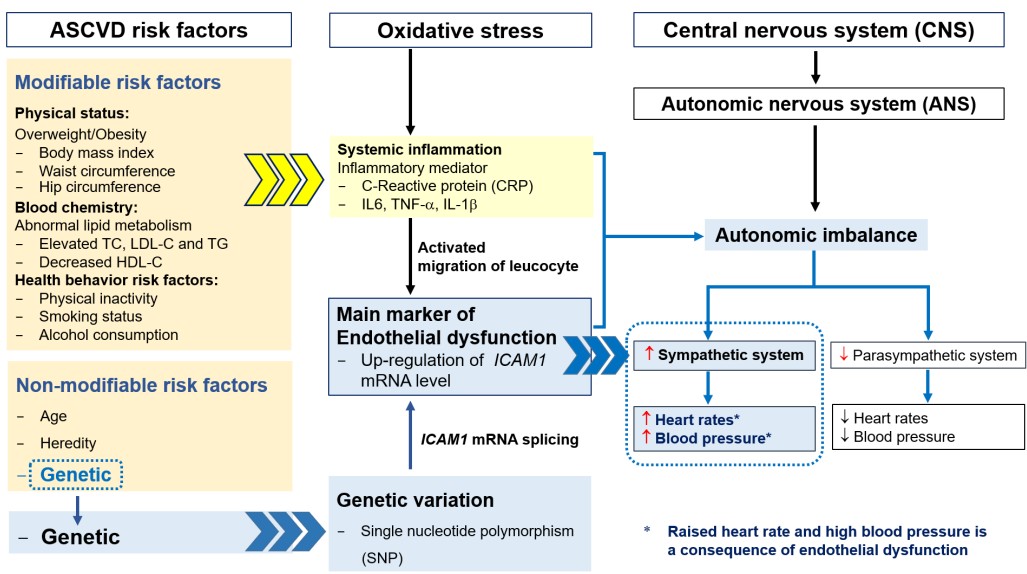

**Figure 3** **The relationships among ASCVD risk factors, oxidative stress, endothelial dysfunction, autonomic imbalance and genetic variation.**

and TG/HDL-C ratio), all of which were high in participants with HCL who were carrying the rs5498 variant. Therefore, we conclude that the rs5498 variant predisposes toward HCL, which may progress to ASCVD.

As mentioned above, high expression of *ICAM1* is a vascular biomarker of endothelial dysfunction. The genetic variants of *ICAM1* affect its expression and promote the development of ASCVD, as shown by numerous studies. In addition, Iwao et al. found that these variants affect RNA splicing: cells with the GG genotype express less *ICAM-1-S* mRNA than those with an AA genotype, and the authors suggested that the g.1548G>A (E469K) polymorphism modifies inflammatory responses by altering cell–cell interactions and regulating apoptosis (*Iwao, Morisaki & Morisaki, 2004*). Similarly, in the present study, we found a significant association between the rs5498 polymorphism and high expression of *ICAM1* mRNA. Therefore, a combination of HCL with the rs5498 variants may promote the development of ASCVD by increasing *ICAM1* expression.

The autonomic nervous system regulates homeostasis and consists of two major branches: the sympathetic and the parasympathetic nervous systems. Autonomic imbalance can result from oxidative stress, systemic inflammation, endothelial dysfunction, and genetic variation, with HCL having an codominant effect to these defects (*Custodis et al., 2010*; *Nanchen et al., 2013a*; *Nanchen et al., 2013b*). The imbalance in the autonomic nervous system is characterized by hyperactivity of the sympathetic system and hypoactivity of the parasympathetic system, which results in high RHR and blood pressure and vascular dysfunction (Fig. 3). High RHR is considered to be a risk factor for heart disease, and can reflect several types of arrhythmia, such as supraventricular tachycardia, atrial fibrillation, sinus tachycardia, and ventricular tachycardia (*Al-Khatib et al., 2018*; *Böhm et al., 2015*; *Page et al., 2016*).

*Thayer, Yamamoto & Brosschot (2010)* suggested that such autonomic imbalance might be the final common pathway linking health conditions, including CVD. Therefore, a change in lifestyle that ameliorates biological risk factors and reduces this autonomic imbalance may reduce the risk of vascular disease (*Christensen et al., 1999*; *Karason et al., 1999*; *Kupari et al., 1993*; *Schroeder et al., 2003*; *Thayer, Yamamoto & Brosschot, 2010*). *Böhm et al. (2015)* also suggested that heart rate is associated with cardiovascular outcomes and other conditions, including endothelial dysfunction. In addition, Nanchen et al. reported an association between RHR and incident heart failure in a population-based cohort study of healthy adults without pre-existing overt heart disease. They suggested that the risk of heart failure increases in men with each increment of 10 beats per minute in RHR (*Nanchen et al., 2013a*; *Nanchen et al., 2013b*). Likewise, we have shown an association between other ASCVD risk factors and a high RHR: after adjusted confounding factor, the rs5498 variant was associated with a 1.58-fold higher risk of ASCVD than reference allele when RHR was ≥ 86 beats/min. Furthermore, the *ICAM1* rs5498 variant was associated with a 5.02-fold higher risk in the presence of tachycardia (≥100 beats/min). This is consistent with systemic inflammation, oxidative stress, and endothelial dysfunction worsening in individuals with HCL and the *ICAM1* variant as heart rate rises. However, we did not identify an association between rs281432 (C>G) and HCL. In contrast, and similar to the present findings, *Yang et al. (2014)* found no significant difference in the prevalences of the rs281432 genotypes in patients with or without coronary atherosclerosis.

The strengths of the present study were that we studied the effect of key genetic variants in a population with the same race, age group, lifestyle, and environment. The sequencing technique we used was able to precisely identify the SNPs. Moreover, we were able to confirm the association between *ICAM1* expression and endothelial dysfunction in ASCVD. However, the cross-sectional design was a limitation, such that we could not show a cause-and-effect relationship between the *ICAM1* rs5498 polymorphism, HCL, and ASCVD.

## CONCLUSIONS

We have shown relationships between the *ICAM1* rs5498 polymorphism and HCL and the related ASCVD risk factors in Thai adults. The underlying mechanisms for this association will involve interactions between abnormal lipid profile, oxidative stress, inflammatory infiltration, and circulating cell adhesion molecules. Although previous studies have indicated that the identification of the *ICAM1* gene variant might be predictive of atherosclerosis in CVD patients, here we show that the variant may predict HCL and therefore may be used in the prevention of ASCVD.

## ACKNOWLEDGEMENTS

We appreciatively acknowledge the nurses at Nong Waeng Health Promoting Hospital for assisting in collecting demographic data and blood samples. We also thank the collaboration of all participants in Sung Noen District, Nakhon Ratchasima Province, Thailand.

### Funding

The funding for this work was supported by the Faculty of Tropical Medicine and the Young Researcher Development Program 2019 of the National Research Council of Thailand (NRCT). The funders had no role in study design, data collection and analysis, decision to publish, or preparation of the manuscript.

### Grant Disclosures

The following grant information was disclosed by the authors:
Faculty of Tropical Medicine and the Young Researcher Development Program 2019 of the National Research Council of Thailand.

### Competing Interests

The authors declare there are no competing interests.

### Author Contributions

- Naruemon Wechjakwen and Karunee Kwanbunjan conceived and designed the experiments, performed the experiments, analyzed the data, prepared figures and/or tables, authored or reviewed drafts of the paper, and approved the final draft.
- Amornrat Aroonnual, Pattaneeya Prangthip and Ngamphol Soonthornworasiri conceived and designed the experiments, analyzed the data, authored or reviewed drafts of the paper, and approved the final draft.
- Pornpimol Panprathip Phienluphon and Jirayu Lainampetch performed the experiments, authored or reviewed drafts of the paper, and approved the final draft.

### Human Ethics

The following information was supplied relating to ethical approvals (i.e., approving body and any reference numbers):

The Ethics Committee of the Faculty of Tropical Medicine (TMEC 18-026), Mahidol University

### DNA Deposition

The following information was supplied regarding the deposition of DNA sequences:

The Homo sapiens ICAM1, RefSeqGene on chromosome 19 are available at GenBank: NG_012083.

### Data Availability

The raw measurements are available in the Supplementary Files.

### Supplemental Information

Supplemental information for this article can be found online at http://dx.doi.org/10.7717/peerj.12972#supplemental-information.

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

# PeerJ

**Expert Panel on Detection, Evaluation, and Treatment of High Blood Cholesterol in Adults. 2001.** Executive summary of the third report of the National Cholesterol Education Program (NCEP) expert panel on detection, evaluation, and treatment of high blood cholesterol in adults (Adult Treatment Panel III). *JAMA* **285**(19):2486–2497 DOI 10.1001/jama.285.19.2486.

**Fox K, Borer JS, Camm AJ, Danchin N, Ferrari R, Lopez Sendon JL, Steg PG, Tardif JC, Tavazzi L, Tendera M. 2007.** Resting heart rate in cardiovascular disease. *Journal of the American College of Cardiology* **50**(9):823–830 DOI 10.1016/j.jacc.2007.04.079.

**Friedewald WT, Levy RI, Fredrickson DS. 1972.** Estimation of the concentration of low-density lipoprotein cholesterol in plasma, without use of the preparative ultracentrifuge. *Clinical Chemistry* **18**(6):499–502 DOI 10.1093/clinchem/18.6.499.

**Gaetani E, Flex A, Pola R, Papaleo P, De Martini D, Pola E, Aloi F, Flore R, Serricchio M, Gasbarrini A, Pola P. 2002.** The K469E polymorphism of the ICAM-1 gene is a risk factor for peripheral arterial occlusive disease. *Blood Coagulation and Fibrinolysis* **13**(6):483–488 DOI 10.1097/00001721-200209000-00002.

**Gazi E, Barutcu A, Altun B, Temiz A, Bekler A, Urfali M, Silan F, Colkesen Y, Ozdemir O. 2014.** Intercellular adhesion molecule-1 K469E and angiotensinogen T207M polymorphisms in coronary slow flow. *Medical Principles and Practice* **23**(4):346–350 DOI 10.1159/000363451.

**Grundy SM, Cleeman JI, Merz CNB, Brewer HB, Clark LT, Hunninghake DB, Pasternak RC, Smith SC, Stone NJ. 2004.** Implications of recent clinical trials for the National Cholesterol Education Program Adult Treatment Panel III guidelines. *Circulation* **110**(2):227–239 DOI 10.1161/01.CIR.0000133317.49796.0E.

**Herbert Haught W, Mansour M, Rothlein R, Kishimoto TK, Mainolfi EA, Hendricks JB, Hendricks C, Mehta JL. 1996.** Alterations in circulating intercellular adhesion molecule-1 and L-selectin: further evidence for chronic inflammation in ischemic heart disease. *American Heart Journal* **132**(1, Part 1):1–8 DOI 10.1016/S0002-8703(96)90383-X.

**Heusch G. 2008.** Heart rate in the pathophysiology of coronary blood flow and myocardial ischaemia: benefit from selective bradycardic agents. *British Journal of Pharmacology* **153**(8):1589–1601 DOI 10.1038/sj.bjp.0707673.

**Iwao M, Morisaki H, Morisaki T. 2004.** Single-nucleotide polymorphism g.1548G > A (E469K) in human ICAM-1 gene affects mRNA splicing pattern and TPA-induced apoptosis. *Biochemical and Biophysical Research Communications* **317**(3):729–735 DOI 10.1016/j.bbrc.2004.03.101.

**Karason K, Mølgaard H, Wikstrand J, Sjöström L. 1999.** Heart rate variability in obesity and the effect of weight loss. *American Journal of Cardiology* **83**(8):1242–1247 DOI 10.1016/S0002-9149(99)00066-1.

**Klop B, Elte JWF, Castro Cabezas M. 2013.** Dyslipidemia in obesity: mechanisms and potential targets. *Nutrients* **5**(4):1218–1240 DOI 10.3390/nu5041218.

**Kupari M, Virolainen J, Koskinen P, Tikkanen MJ. 1993.** Short-term heart rate variability and factors modifying the risk of coronary artery disease in a population sample.

*American Journal of Cardiology* **72(12)**:897–903
DOI 10.1016/0002-9149(93)91103-O.

**Li D, Qu C, Dong P. 2014.** The ICAM-1 K469E polymorphism is associated with the risk of coronary artery disease: a meta-analysis. *Coronary Artery Disease* **25(8)**:665–670 DOI 10.1097/MCA.0000000000000136.

**Linton MRF, Yancey PG, Davies SS, Jerome WG, Linton EF, Song WL, Doran AC, Vickers KC. 2019.** The role of lipids and lipoproteins in atherosclerosis. In: Feingold KR, Anawalt B, Boyce A, Chrousos G, De Herder WW, Dhatariya K, Dungan K, Hershman JM, Hofland J, Kalra S, Kaltsas G, Koch C, Kopp P, Korbonits M, Kovacs CS, Kuohung W, Laferrére B, Levy M, McGee EA, McLachlan R, Morley JE, New M, Purnell J, Sahay R, Singer F, Sperling MA, Stratakis CA, Trence DL, Wilson DP, eds. *Endotext.* South Dartmouth: MDText.com, Inc. 2000. *Available at* www.ncbi.nlm.nih.gov/sites/books/NBK343489.

**Liu LZ, Wu EP, Liu HL. 2013.** Relation between K469E gene polymorphism of ICAM-1 and recurrence of ACS and cardiovascular mortality. *Asian Pacific Journal of Tropical Medicine* **6(11)**:916–920 DOI 10.1016/S1995-7645(13)60164-9.

**Mach F, Baigent C, Catapano AL, Koskinas KC, Casula M, Badimon L, Chapman MJ, De Backer GG, Delgado V, Ference BA, Graham IM, Halliday A, Landmesser U, Mihaylova B, Pederson TR, Riccardi G, Richter DJ, Sabatine MS, Taskinen MR, Tokgozoglu L, Wiklund O. ESC Scientific Document Group. 2019.** 2019 ESC/EAS Guidelines for the management of dyslipidaemias: lipid modification to reduce cardiovascular risk: the task force for the management of dyslipidaemias of the European Society of Cardiology (ESC) and European Atherosclerosis Society (EAS). *European Heart Journal* **41(1)**:111–188 DOI 10.1093/eurheartj/ehz455.

**Manktelow BN, Potter JF. 2009.** Interventions in the management of serum lipids for preventing stroke recurrence. *Cochrane Database of Systematic Reviews* **2009(3)**:1–22 DOI 10.1002/14651858.CD002091.pub2.

**Martinez-Hervas S, Ascaso JF. 2019.** Hypercholesterolemia. In: Huhtaniemi I, Martini L, eds. *Encyclopedia of endocrine diseases*. Second Edition. Oxford: Academic Press, 320–326.

**Nanchen D, Leening MJG, Locatelli I, Cornuz J, Kors JA, Heeringa J, Deckers JW, Hofman A, Franco OH, Stricker BHCH, Witteman JCM, Dehghan A. 2013a.** Resting heart rate and the risk of heart failure in healthy adults. *Circulation: Heart Failure* **6(3)**:403–410 DOI 10.1161/CIRCHEARTFAILURE.112.000171.

**Nanchen D, Stott DJ, Gussekloo J, Mooijaart SP, Westendorp RG, Jukema JW, Macfarlane PW, Cornuz J, Rodondi N, Buckley BM, Ford I, Sattar N, De Craen AJ. 2013b.** Resting heart rate and incident heart failure and cardiovascular mortality in older adults: role of inflammation and endothelial dysfunction: the PROSPER study. *European Journal of Heart Failure* **15(5)**:581–588 DOI 10.1093/eurjhf/hfs195.

**Nasibullin TR, Timasheva YR, Sadikova RI, Tuktarova IA, Erdman VV, Nikolaeva IE, Sabo J, Kruzliak P, Mustafina OE. 2016.** Genotype/allelic combinations as potential predictors of myocardial infarction. *Molecular Biology Reports* **43(1)**:11–16 DOI 10.1007/s11033-015-3933-3.

**Nepal G, Yadav JK, Kong Y. 2019.** Association between K469E polymorphism of ICAM-1 gene and susceptibility of ischemic stroke: an updated meta-analysis. *Molecular Genetics and Genomic Medicine* **7**(7):e00784 DOI 10.1002/mgg3.784.

**O'Gara PT, Kushner FG, Ascheim DD, Casey DE, Chung MK, De Lemos JA, Ettinger SM, Fang JC, Fesmire FM, Franklin BA, Granger CB, Krumholz HM, Linderbaum JA, Morrow DA, Kristin Newby L, Ornato JP, Ou N, Radford MJ, Tamis-Holland JE, Tommaso CL, Tracy CM, Joseph Woo Y, Zhao DX. 2013.** 2013 ACCF/AHA Guideline for the management of ST-Elevation myocardial infarction. *Circulation* **127**(4):e362–e425 DOI 10.1161/CIR.0b013e3182742cf6.

**Page RL, Joglar JA, Caldwell MA, Calkins H, Conti JB, Deal BJ, Mark Estes NA, Field ME, Goldberger ZD, Hammill SC, Indik JH, Lindsay BD, Olshansky B, Russo AM, Shen WK, Tracy CM, Al-Khatib SM. 2016.** 2015 ACC/AHA/HRS Guideline for the management of adult patients with supraventricular tachycardia. *Circulation* **133**(14):e506-e574 DOI 10.1161/CIR.0000000000000311.

**Palatini P. 2007.** Heart rate as an independent risk factor for cardiovascular disease: current evidence and basic mechanisms. *Drugs* **67**(**Supp 2**):3–13 DOI 10.2165/00003495-200767002-00002.

**Reil JC, Böhm M. 2007.** The role of heart rate in the development of cardiovascular disease. *Clinical Research in Cardiology* **96**(9):585–592 DOI 10.1007/s00392-007-0537-5.

**Sarecka-Hujar B, Zak I, Krauze J. 2009.** Interactions between rs5498 polymorphism in the ICAM1 gene and traditional risk factors influence susceptibility to coronary artery disease. *Clinical and Experimental Medicine* **9**(2):117–124 DOI 10.1007/s10238-008-0022-0.

**Schroeder EB, Liao D, Chambless LE, Prineas RJ, Evans GW, Heiss G. 2003.** Hypertension, blood pressure, and heart rate variability. *Hypertension* **42**(6):1106–1111 DOI 10.1161/01.HYP.0000100444.71069.73.

**Shaker O, Zahra A, Sayed A, Refaat A, El-Khaiat Z, Hegazy G, El-Hindawi K, Deen MAE. 2010.** Role of ICAM-1 and E-selectin gene polymorphisms in pathogenesis of PAOD in Egyptian patients. *Vascular Health and Risk Management* **6**:9–15 DOI 10.2147/vhrm.s8143.

**Sonja S, Milica M, Slađana S. 2017.** Biomarkers of endothelial dysfunction in cardiovascular diseases. *Medicinski Pregled* **70**(1-2):53–57 DOI 10.2298/MPNS1702053S.

**Thayer JF, Yamamoto SS, Brosschot JF. 2010.** The relationship of autonomic imbalance, heart rate variability and cardiovascular disease risk factors. *International Journal of Cardiology* **141**(2):122–131 DOI 10.1016/j.ijcard.2009.09.543.

**Van Dijk SJ, Tellam RL, Morrison JL, Muhlhausler BS, Molloy PL. 2015.** Recent developments on the role of epigenetics in obesity and metabolic disease. *Clinical Epigenetics* **7**(1):66 DOI 10.1186/s13148-015-0101-5.

**Verbeek R, Hoogeveen RM, Langsted A, Stiekema LCA, Verweij SL, Hovingh GK, Wareham NJ, Khaw KT, Boekholdt SM, Nordestgaard BG, Stroes ESG. 2018.** Cardiovascular disease risk associated with elevated lipoprotein(a) attenuates at low

low-density lipoprotein cholesterol levels in a primary prevention setting. *European Heart Journal* **39**(27):2589–2596 DOI 10.1093/eurheartj/ehy334.

**Virani SS, Alonso A, Benjamin EJ, Bittencourt MS, Callaway CW, Carson AP, Chamberlain AM, Chang AR, Cheng S, Delling FN, Djousse L, Elkind MSV, Ferguson JF, Fornage M, Khan SS, Kissela BM, Knutson KL, Kwan TW, Lackland DT, Lewis TT, Lichtman JH, Longenecker CT, Loop MS, Lutsey PL, Martin SS, Matsushita K, Moran AE, Mussolino ME, Perak AM, Rosamond WD, Roth GA, Sampson UKA, Satou GM, Schroeder EB, Shah SH, Shay CM, Spartano NL, Stokes A, Tirschwell DL, Van Wagner LB, Tsao CW. 2020.** Heart disease and stroke statistics-2020 update: a report from the American Heart Association. *Circulation* **141**(9):e139–e596 DOI 10.1161/cir.0000000000000757.

**Wang D, Zhang FH, Zhao YT, Xiao XG, Liu S, Shi HB, Lin AL, Wang YJ, Han Q, Sun QM. 2015.** Association of polymorphism in ICAM-1 (K469E) and cytology parameters in patients' initial blood test with acute ischemic stroke. *Genetics and Molecular Research* **14**(4):15520–15529 DOI 10.4238/2015.December.1.2.

**World Health Organization (WHO). 2000.** *The Asia-Pacific perspective: redefining obesity and its treatment.* Sydney: Health Communications Australia.

**World Health Organization (WHO). 2004.** Appropriate body-mass index for Asian populations and its implications for policy and intervention strategies. *The Lancet* **363**(9403):157–163 DOI 10.1016/S0140-6736(03)15268-3.

**Yang M, Fu Z, Zhang Q, Xin Y, Chen Y, Tian Y. 2014.** Association between the polymorphisms in intercellular adhesion molecule-1 and the risk of coronary atherosclerosis: a case-controlled study. *PLOS ONE* **9**(10):e109658 DOI 10.1371/journal.pone.0109658.

**Yin DL, Zhao XH, Zhou Y, Wang Y, Duan P, Li QX, Xiong Z, Zhang YY, Chen Y, He H, Yang K, Song HJ. 2019.** Association between the ICAM-1 gene polymorphism and coronary heart disease risk: a meta-analysis. *Bioscience Reports* **39**(2):BSR20180923 DOI 10.1042/bsr20180923.