# Peer review of "Associations between the rs5498 (A > G) and rs281432 (C > G) polymorphisms of the ICAM1 gene and atherosclerotic cardiovascular disease risk, including hypercholesterolemia"

_PeerJ, doi:10.7717/peerj.12972_

## Round 0.1 · original submission · Major Revisions

After analyzing the reviewers' reports, I think your manuscript can be considered for publication in PeerJ. However, there are some modifications needed which are relevant for improving your work in a revised version. Please, see the reviewers' comments to have more information.

Reviewer 1 ·

Basic reporting

Here, they reported that the G allele in ICAM1 rs5498 is linked to increased ICAM1 mRNA expression, LDL-C concentration, and RHR, which might also lead to increased ASCVD risk and HCL in Thai adults.

Experimental design

ICAM1 SNPs Genotyping was performed in normal controls and HCL individuals assessed by clinical characteristics.

Validity of the findings

This prevalence study analyzing data from 278 participants in Thailand and showed the relationships between the ICAM1 rs5498 (A>G) polymorphism and hypercholesterolemia (HCL) and the atherosclerotic cardiovascular disease (ASCVD) risk factors in them.

Additional comments

(1) Although the data shown in this study suggested that ICAM1 rs5498 (A>G) might be linked to HCL in ASCVD, its effect on increased ICAM1 mRNA level was not significant (Table 1 and Table 3). Only in Fig. 2, normal control with wild type AA showed the significantly less expression of ICAM mRNA than HCL group with variant (AG+GG).
(2) In this study, it was not clear why the G allele in ICAM1 rs5498 is closely linked to increased the risk for ASCVD and HCL in participants.
(3) Does increased ICAM1 expression lead to the increased heart rate or increased sympathetic activation?

Annotated reviews are not available for download in order to protect the identity of reviewers who chose to remain anonymous.

Reviewer 2 ·

Basic reporting

Wechjakwen and colleagues investigate the role of two polymorphisms of ICAM gene on hypercholesterolemia and atherosclerotic cardiovascular disease. Despite the research topic being relevant, this study has some flaws and it is not completely clear what new message the authors want to bring in comparison with the other studies of the field.
Hence the authors should make modifications to make the manuscript clearer.

1. In the abstract the risk factors that are going to be taken in consideration should be introduced, to have an overview of the analyses performed.

2. Study design and participants: How subjects where recruited?
3. Study design and participants: How enviromental variables were assessed?
4. Line 115: normal control group should be defined as control only. And wild-type is not completely appropriate when we are talking about genotype. It's reference allele the correct definition.
5. Table 4. a. the same analysis should be shown also for rs281432. b. P-value for Chi-squared test should be reported for the difference between low-risk and high risk c. c.The part of mRNA expression should be moved separately. d. Is there some reason why blood pressore was not included?
6.Discussion. The authors should be more cautios in saying that they demonstrate the association of the snp with ASCVD. They just reported the association with some of the risk factor separately.

Experimental design

The techniques used were correct.

Validity of the findings

There would need further investigation including inflammatory markers, to validate the analyses.

---

## Round 0.2 · accepted · Accept

All the previous concerns have been correctly addressed, therefore I recommend the publication of your work in PeerJ. Congratulations!

Reviewer 2 ·

Basic reporting

The authors provided improvements to the manuscript, for all the issues proposed in the previous review process.

Experimental design

The authors provided improved the manuscript for all the issues proposed in the previous review process.

Validity of the findings

The authors provided improved the manuscript for all the issues proposed in the previous review process.